# An Ensemble Model for the Diagnosis of Brain Tumors through MRIs

**DOI:** 10.3390/diagnostics13030561

**Published:** 2023-02-03

**Authors:** Ehsan Ghafourian, Farshad Samadifam, Heidar Fadavian, Peren Jerfi Canatalay, AmirReza Tajally, Sittiporn Channumsin

**Affiliations:** 1Department of Computer Science, Iowa State University, Ames, IA 50010, USA; 2Department of Biomedical Engineering, Medical College of Wisconsin, Milwaukee, WI 53226, USA; 3Department of Electrical and Computer Engineering, Tarbiat Modares University, Tehran 14115-111, Iran; 4Computer Engineering, Faculty of Engineering Department, Halic University, Istanbul 34394, Turkey; 5School of Industrial Engineering, College of Engineering, University of Tehran, Tehran 14179-35840, Iran; 6Space Technology Research Center, Geo-Informatics and Space Technology Development Agency (GISTDA), Chonburi 20230, Thailand

**Keywords:** machine learning, magnetic resonance imaging, ensemble classifier, singular value decomposition, social spider optimization

## Abstract

Automatic brain tumor detection in MR Images is one of the basic applications of machine vision in medical image processing, which, despite much research, still needs further development. Using multiple machine learning techniques as an ensemble system is one of the solutions that can be effective in achieving this goal. In this paper, a novel method for diagnosing brain tumors by combining data mining and machine learning techniques has been proposed. In the proposed method, each image is initially pre-processed to eliminate its background region and identify brain tissue. The Social Spider Optimization (SSO) algorithm is then utilized to segment the MRI Images. The MRI Images segmentation allows for a more precise identification of the tumor region in the image. In the next step, the distinctive features of the image are extracted using the SVD technique. In addition to removing redundant information, this strategy boosts the speed of the processing at the classification stage. Finally, a combination of the algorithms Naïve Bayes, Support vector machine and K-nearest neighbor is used to classify the extracted features and detect brain tumors. Each of the three algorithms performs feature classification individually, and the final output of the proposed model is created by integrating the three independent outputs and voting the results. The results indicate that the proposed method can diagnose brain tumors in the BRATS 2014 dataset with an average accuracy of 98.61%, sensitivity of 95.79% and specificity of 99.71%. Additionally, the proposed method could diagnose brain tumors in the BTD20 database with an average accuracy of 99.13%, sensitivity of 99% and specificity of 99.26%. These results show a significant improvement compared to previous efforts. The findings confirm that using the image segmentation technique, as well as the ensemble learning, is effective in improving the efficiency of the proposed method.

## 1. Introduction

A brain tumor is a hard, solid neoplasm that develops within the brain or the spinal cord’s central canal. Ordinarily, a brain tumor is an abnormal mass in the brain that can be cancerous (malignant) or benign (noncancerous). The threat level of a tumor is determined by a number of factors, including its type, location, size, life expectancy, and how it spreads and develops. The skull completely encases the brain. Only the presence of paraclinical tools and appropriate diagnostic tools to assess the condition of the intracranial cavity in the early stages of tumor formation allows the rapid and early diagnosis of brain tumors [1]. Even with these tools, brain tumors are extremely difficult to diagnose due to their wide range of shapes, sizes, and appearances. Brain tumors are one of the most common types of cancer, accounting for a significant number of deaths each year. Accurate and timely diagnosis of this complication can help treat the disease and prevent its progression [2]. At present, the diagnosis of the tumor area in brain images is made by physicians and radiologists. Factors such as low experience, fatigue, or lack of concentration can lead to misdiagnosis. Automated techniques based on machine learning can be effective in reducing the risk of error in such situations. As a result of this, and considering the rising prevalence of brain diseases, such as cancerous tumors, many researchers have attempted to detect this disease early. With the advancement of computer-based technologies in recent years, the use of these technologies in detecting tumors has increased. Machine Learning (ML) [3] is one of the computer methods that have been considered for detecting brain tumors. The research history about brain tumor diagnosis includes various ML-based methods. This research has attempted to improve diagnosis efficiency by better pre-processing or more efficient feature extraction. Among these, researchers have paid less attention to approaches such as combining classifiers for diagnosis, and most of them have focused on comparing rather than combining. Meanwhile, the combination of ML models can have a significant effect in improving the accuracy of diagnosis. This has motivated the authors to conduct the current research.

In this paper, machine learning techniques are used to provide an efficient method for diagnosing brain tumors. The proposed method includes four main steps: Preprocessing, segmentation, feature extraction, and classification. The contribution of this research is twofold:Firstly, the proposed method uses the Social Spider Optimization (SSO) algorithm for the segmentation of brain MRIs. This operation allows for a more precise identification of the tumor region in the image. Additionally, the segmentation of images reduces the problem of complexity by limiting the pixel values and discarding redundant details in the image. The Singular Value Decomposition (SVD) technique is also used to extract the image characteristics. Both these techniques will lead to increasing the detection accuracy and processing speed in the classification phase.Secondly, the proposed method uses a combination of three classifiers as an ensemble model for diagnosing brain tumors. These classification models include: Naive Bayes (NB), Support Vector Machine (SVM), and K Nearest Neighbor (KNN). Using ensemble learning is effective in reducing the diagnosing error, because each pair of classifiers may cover the error of the other one through the voting technique.

The above cases have not been addressed in previous research concerning brain tumor diagnosis, and can be considered as innovations of the research. The following details how the rest of the paper is organized: the background research is reviewed in the Section 2 and the proposed method for diagnosing brain tumors in MRIs is presented in the Section 3. The results of the implementation and an evaluation of the proposed method are presented in the Section 4, and the results are discussed in the Section 5. Finally, in the Section 6, the conclusions are made and several solutions for further research have been suggested.

## 2. Related Works

Because of the subject’s importance in recent years, a lot of research has been conducted in the field of diagnosing various diseases related to the human brain, including research about the diagnosis of brain tumors using MR Images. In this section, several studies on brain tumor diagnosis are highlighted. In [4], Shobana and Balakrishnan proposed a method for classifying brain tumors by applying Discrete Cosine Transform (DCT) and Discrete Wavelet Transform (DWT), followed by a Probablistic Neural Network (PNN). They also used entropy and image energy as features. Shanmagapria and Ramakrishnan [5] used a SVM to classify brain tumors in images, and after examining two different kernels of the support vector machine, they concluded that taking into account density and severity characteristics can help improve diagnostic accuracy. They also demonstrated that using a SVM with a Radial Basis Function (RBF) kernel improves accuracy. Dashban et al. [6] used fuzzy clustering techniques and a SVM to diagnose brain tumors on MR Images. A SVM was used to diagnose a brain tumor in the study [7]. Different kernel functions were used to classify the data in this paper, in which the quadratic kernel function showed the best accuracy.

Deep learning techniques have recently been used widely for medical image processing purposes [8]. Deep learning models cover a vast range of medical applications such as diagnosis of gastroscopy biopsy [9], lung cancer detection [10] and medical images similarity retrieval [11]. These methods have also been widely used in brain tumor diagnosis. A deep learning segmentation method for brain tumor diagnosis is proposed in [12]. In this method, a deep neural network is used to extract features from MRIs. This model can process large-volume MRIs quickly and accurately. A Convolutional Neural Network (CNN) technique is also used in [13] to detect tumors in brain MR images. In this method, a CNN optimized by the bat search algorithm is used. To detect brain tumors in MRIs, the method presented in [14] used a combination of CNN and the Whale Harris Hawks Optimization (WHHO) algorithm.

In [15], an ensemble of deep features and machine learning algorithms were used for brain tumor diagnosis in MRIs. In this research, 13 pre-trained CNN models have been used for extracting deep features from MRIs. Then, 9 classifiers were used for selecting top-3 deep features. In the next step, an ensemble of the selected features (through concatenation) were fed to a SVM with RBF kernel function. Authors in [16] have focused on enhancing MRIs in order to improve the brain tumor classification accuracy. Their method included three main phases: noise removal using median filter, contrast enhancement using Histogram Equalization (HE), and image conversion from grayscale to RBG. The authors have used a DNN for classification. In [17], an automatic method for brain tumor segmentation has been proposed, which uses multi-scale residual attention-UNet (MRA-UNet). This deep learning model is fed with three consequent slices of MRI for preserving the sequential information.

In [18], transfer learning was used to extract features from MRIs using a CNN. In this method, first three isolated CNNs with multiple layers are trained to evaluate the performance. Then, the developed CNNs are used for extracting features and training a SVM through transfer learning. In [19], ensemble deep learning has been used for brain tumor diagnosis. This ensemble model includes two deep learning algorithms: CNN and LSTM. Authors in [20] have developed a framework for brain tumor diagnosis using isolated and transferred deep learning. In this method, various CNNs are trained for checking performance. Then, a binary classifier including a CNN with 22 layers is reused for classifying brain MRI images into tumor subclasses.

Authors in [21] proposed a method for brain tumor diagnosis through MRIs using orthogonal wavelet transform and deep learning techniques. In this method, input images are pre-processed and then the significant features of each image are described by a combination of statistical features and gray-level-co-occurrence matrix (GLCM). A genetics Algorithm (GA) is also used for features selection. Finally, the selected features are fed to a CNN for classification. In [22], a new method based on the combination of the GA and U-Net neural network has been proposed for diagnosing brain tumors. In this method, GA is used for image segmentation. Then, features of the segmented image are extracted using discrete wavelet transform and a subset of these features is selected using particle swarm optimization. Finally, a U-Net is used for the classification of features. In [23], a hybrid method based on CNN and SVM with RBF kernel has been proposed for diagnosing brain tumors in MRIs. This method uses three subsequent CNN models for diagnosis. The first CNN is used to study the feature map from MRIs. The second CNN, is a fast region-based model for tumor localization. The third CNN is combined with a SVM model for classification of region features and diagnosing brain tumor.

Research in [24] describes the social spider optimization algorithm, which is a swarm intelligence algorithm. The main driver of the search engine in SSO is artificial spider mites. Each spider has a memory consisting of the current position of the web, the amount of current position fitness, and the vibration (position and intensity) in the previous iteration. The first two parts of the information describe the features of this spider, and the last feature helps in guiding the spider’s motion algorithm. The spider detects very sensitive vibrations, according to biological observations. It can detect the strength and direction of vibrations with great accuracy, and it can create multiple, even separate vibrations within the same network fiber. SSO is used to take advantage of this feature of the spider and create information communication between the spiders based on the vibration of the system. When a spider moves to different positions and previous positions in SSO, it generates a new vibration. The vibration of the spider is then transmitted through the web and felt by others. This means that the spider shares its personal information in a social manner. The position of the source and the intensity (quality) of the source are the two characteristics that define vibrations in SSO. Vibration is generated at the current position, and then, the spider moves to a new position.

Initialization, iteration, and solution determination are the three phases of SSO. The fitness function, search bounds, and SSO optimization parameters are all initialized during the initialization phase of each SSO implementation. The spider web is then filled with a randomly generated population of spiders. Each spider’s position in the search space is generated randomly, and the vibration intensity is zero. After this initial phase, we move on to the SSO iteration phase. The optimization operation is performed iteratively during the iteration phase of the algorithm. There is a predefined number of iterations. The algorithm calculates the number of spiders that can fit on the web in each iteration and then reduces vibrations in comparison to the previous iteration. Then, in its current position, each spider produces vibrations. After that, the vibrations travel through the web strings. After being released, each spider will experience a variety of vibrations coming from all directions. After receiving these weak vibrations, each spider chooses the strongest vibration, such as the strongest best vibration. Following these weak vibrations, each spider selects the most intense vibration, such as *v_best_*. The vibrations stored by the spider in the previous iteration, *v_prev_*, are then compared to the *v_best_*, and the spider moves toward the stronger vibration. The loop iteration step is repeated until all of the criteria have been met. The best solution with the desired fitness value is the algorithm’s output [24]. The studied works are summarized in Table 1.

## 3. Research Method

In this section, we will describe the proposed method for the diagnosis of brain tumors using the ensemble technique in detail. This research is an attempt to improve brain tumor diagnosis compared to previous studies. Most of the previous studies use a single classifier for diagnosing brain tumors in MRIs, which leads to a relatively high error rate. The proposed method combines three classifiers as an ensemble system for this task. In this case, multiple classifiers are able to cover the errors of each one. Thus, the ensemble learning may be effective in increasing the accuracy of diagnosis. Figure 1 shows a block diagram of the proposed method. 

The proposed model, as shown in Figure 1, includes an MRI database as well as components for preprocessing, segmentation, feature extraction, and classification. The proposed method involves the following steps for diagnosing brain tumors using MRIs:Image preprocessing: Regions of the brain are identified in this step, and the results are used as input to the next steps of the proposed method.Image feature description: For segmentation of MRIs, the social spider optimization algorithm and multilevel thresholding technique will be used.Feature extraction: In this step, the dimensions of image features are reduced using the SVD algorithm, so that, in addition to increasing the processing speed of the proposed method, irrelevant features are removed and the detection operation is performed with the highest accuracy possible.Classification of extracted features: The extracted features will be classified using an ensemble model and a combination of SVM, Naive Bayes, and KNN, in this step. The goal of using ensemble techniques in the diagnosis of brain tumors is to reduce the error of classification algorithms (see Figure 1).

The proposed model, deals with the over-fitting problem by means of feature extraction and ensemble learning techniques. The SVD algorithm in the feature extraction step, is effective in data simplification and reduces redundant features to form less complex learners. On the other hand, the ensemble learning mechanism combines several classifiers by training them in a parallel manner to form a stronger diagnosis system that is more robust in over-fitting conditions. A reference table for the notations used in this paper is presented Table 2.

### 3.1. MRI Preprocessing

The goal of the preprocessing step is to refine MRIs by removing their background. This operation helps to get rid of any redundant data in the image, as well as any data that might interfere with the detection process. We first convert the input image to binary using the 0.05 threshold value to remove the background. The threshold value of 0.05 in a binary image is equivalent to the intensity value of $0.05\times \left(2^{8}-1\right)=12.75$ in an 8-bit grayscale image. The background area in brain MRI images has an intensity lower than 12 and the skull and brain areas have an intensity higher than this value; therefore, it can be ensured that this threshold value can approximately separate the brain borders from the background area in MRIs. Each pixel in row *i* and column *j* of the input image *I* is converted to a part in the binary image *B* using the following equation:(1)$$B_{i,j}=\left\{\begin{array}{ll} 0 & \;if\;I_{i,j}\leq 0.05\times 255\\ 1 & \;if\;I_{i,j}>0.05\times 255 \end{array}\right.$$

After generating a binary image *B*, we find the largest connected area in image *B* that has the value of 1. The resulting connected region may not be integrated and contains holes with zero values. We fill the holes in the selected area with values of 1 and set all the points that are not members of the selected area to zero due to the integration of the foreground region in MRIs. This will result in the creation of a binary image. We use the following equation to obtain the foreground image:*FG* = *B*′·*I*(2)

The background part of the image will be removed by multiplying each pixel in the image *I* by the corresponding parts in image *B*. The segmentation algorithm will use the resulting preprocessed image as the input. This algorithm will be described in the following sections.

### 3.2. Segmentation of Images Using SSO Algorithm

As previously stated, the SSO algorithm will be used to segment the preprocessed images in this step. The SSO algorithm is an efficient method that, compared to other optimization algorithms (such as GA or PSO), does not need to consider various parameters. Additionally, the multilevel thresholding problem can easily be mapped to an optimization problem which can be solved by SSO. These features have made the SSO algorithm a suitable solution for the purpose of MRI segmentation. The goal of the segmentation algorithm is to separate the suspected lesions in the image from other regions of the brain. An MRI can be large in size and therefore the number of pixels in it will be very large. On the other hand, each pixel can have a value between zero and 255. Because of the large size of MRIs and the wide range of possible values for each pixel, distinguishing these pixels and detecting the tumor region in them is difficult. Limiting the problem space is one solution to this problem. It is necessary to be able to remove unnecessary details from an image and perform a more accurate and faster detection operation. This can be accomplished in two methods:Reducing the number of pixels.Limiting the range of possible values for each pixel in the image.

The first option improves the processing speed, but it may also remove some useful data. Therefore, reducing the number of pixels should be considered based on image properties. The second solution, on the other hand, can be accomplished through segmentation operations. This will provide two benefits. The first benefit of MRI segmentation is that it restricts the range of values that can be assigned to each pixel in the image. As a result, the pixel value range reduces from [0, 255] to [1, *C*] (the parameter C≪255 is set by the user and specifies the number of segments). The second benefit of segmentation is that it allows for a more comprehensive understanding of pixel correlation features. The pixels corresponding to the same regions in a segmented image have the same values. Thus, areas of a suspected lesion can be identified using segmentation.

The dimensions of the input image are reduced using the averaging technique to improve system performance and keep the segmentation speed from slowing down. The average value of each of the four pixels adjacent to each other in a 2 × 2 matrix is treated as one image pixel in this technique. After reducing the dimensions of the input image, it is converted to a matrix such as $I_{W\times H}$ and then segmented to *c* regions using the SSO algorithm. The search mechanism in the social spider optimization algorithm is detailed in [24], and the same process is used to segment the images in the proposed method. Assuming the reader is familiar with this optimization algorithm, describing the solution vector structure and fitness function will suffice. Each solution in the proposed method specifies the image segmentation thresholds. Figure 2 shows the structure of a solution vector and how to segment images using the SSO algorithm.

In the example shown in Figure 2, the number of image segments is considered as four. As a result, the solution vector will have a length of four. The first value in the solution vector, for example, is 20. As a result, all pixels with a value in the range $\left[0,\;20\right]$ are segmented in the first region. This region covers the image’s background (in Figure 2, the member pixels of each region are shown in white). In the same way, the second value in the solution vector is 80. As a result, the second region contains all pixels with a value in the range $\left[21,\;80\right]$. The final segmented image will be constructed by combining all of the created regions. The output of this operation is depicted in the lower section of Figure 2. Each area in this diagram is represented by a different color. The image segmentation parameters to be optimized are multilevel thresholding values. As a result, the number of thresholding values equals the length of each solution vector. Each of the previously mentioned optimization parameters can be used to calculate the best solution as a natural number in the interval $\left[0,\;255\right]$. In the proposed method, the optimization algorithm’s initial population is generated randomly.

The fitness function is the most important part of an optimization algorithm. The optimality of the solution is described by a fitness function. As a result, the fitness function can be used to determine which region for each image pixel is more suitable and which solution in the search algorithm is the best. When considering an image such as *X*, each pixel *I_x,y_* in this image can be assigned to one of the *m* target regions *T_i_*, using the threshold values of $\left\{t_{1},\ldots ,t_{m}\right\}$:*T*_0_ = {*Ix,y* ∈ *X* | 0 ≤ *Ix,y* ≤ *t*_1_ − 1};
*T*_1_ = { *Ix,y* ∈ *X* | *t*_1_ ≤ *Ix,y* ≤ *t*_2_ − 1}
*T_i_* = { *Ix,y* ∈ *X* | *t_i_* ≤ *Ix,y* ≤ *t_i_*_+1_ − 1}
*T_m_* = { *Ix,y* ∈ *X* | *t_m_* ≤ *Ix,y* ≤ *255*}(3)
where *X* is the input image for segmentation and *I_x,y_* represents the intensity of the pixel in position *(x,y)*, and *t_i_* represent the *i*-th threshold value for determining the regions on the image. As a result, the input image can be segmented into the *m* regions using these threshold values. In other words, the SSO algorithm has the same number of optimization variables as target thresholds. The proposed method considers different thresholding values for each possible solution in the optimization algorithm and then calculates the solution fitness using the Tsallis entropy criterion [25].
(4)$$S_{q}=\frac{1-\sum_{i=1}^{K}p_{i}^{q}}{q-1}$$
where *p_i_* represents the model’s position in a state *i* and is in the range of zero to one. This parameter corresponds to the number of brightness levels in an MRI with a greyscale color system.

### 3.3. Extracting Features Using SVD

In the third step of the propose method, the SVD algorithm is used to extract the features. Consider a database containing *P* images, where the dimensions of each sample after segmentation are S=A×B. We convert each segmentated image into a vector to extract the feature using the SVD algorithm. As a result, the entire database can be described as $R_{S\times P}$. The SVD algorithm is based on linear algebra theory, which states that a square matrix such as R can be decomposed into three matrices:The orthogonal matrix *U*Diagonal matrix ∑Transpose of the orthogonal matrix *V*

The SVD algorithm’s goal is to obtain a square matrix and compress it into a smaller space. The dimensions of the feature matrix are reduced using this algorithm. After the noise is removed, this technique reveals hidden data. The SVD calculation divides matrix R into three matrices [26]:ℛ = *UΣV^T^*(5)

The columns of *U* are the eigenvectors of matrix $RR^{T}$. This matrix is referred to as the left eigenvector. There is also an orthogonal matrix *∑*, with diagonal elements of singular values of R and non-diagonal elements of zero. The correlation between the features is described in this matrix. Finally, *V* is a matrix whose columns are $R^{T}R$ eigrnvectors. This matrix is known as the right eigenvector matrix. The transpose of *V* is represented by *V^T^*. This matrix is used in the proposed method to rank the features. Gong and Liu demonstrated in [27] that the order of the rows in this matrix indicates the importance of the features in the database; the first line represents the most important feature and so on. On the other hand, it should be possible to specify an appropriate number of features in order to reduce its size. Based on the characteristics of the MR images, the number of suitable features can be determined. According to research, the average effect of noise on a brain MRIs is 1%. As a result, the number of features suitable for brain MRI Images would be the number of features whose sum of squares equals 99 percent of the sum of squares of Σ the original diameter of Σ. If we consider this number of features to be *N*, we can obtain the features extracted by the SVD method using the following equation [26]:(6)$$F_{p\times N}=U_{p\times N}\times \Sigma_{N\times N}\times V_{N\times N}^{T}$$

The above equation shows that the extracted features matrix can be obtained by multiplying the coefficients of the *N* superior features in the decomposition matrices.

### 3.4. Classification and Diagnosis of Brain Tumors Using Ensemble Technique

To diagnose brain tumors, the proposed method employs the ensemble technique and a combination of classification algorithms. In the proposed ensemble system, three learning algorithms are used to classify features and diagnose brain tumors: SVM, Naive Bayes, and KNN.

Although each of the above algorithms have an acceptable performance in classifying samples and diagnosing brain tumors independently, they are still far from an optimal diagnosis system. To bring the performance of the proposed model closer to an optimal tumor detection system, the capabilities of these classifiers can be combined. Thus, in the combination of the three above classifiers, each pair of learning models can cover the error of the third learning model and thus reduce the overall error of the system in diagnosing the problem. In the following sections, we will evaluate how to classify the extracted features using each of these algorithms and then describe how to combine their output in proposed ensemble system.

#### 3.4.1. Support Vector Machine

The SVM is the first learning model used in the proposed method. Two hyperplanes with definite boundaries and positions relative to each other can be used to describe a SVM. Each hyperplane belongs to one of the target classes, and the margin is the shortest distance between the instances of each class and the border of the hyperplane. The SVM is a classification algorithm that attempts to maximize classification accuracy by increasing the margin between each class’s sample holding planes. To find the line that separates the classes, these algorithms begin with two parallel lines that are moved in opposite directions until each line reaches a sample of a specific category on its side. A bar or border is formed between two parallel lines as a result of this step. The wider the bar, the more the algorithm can maximize the margin, which is the goal [28]. To classify the features, the proposed method employs a SVM with a linear kernel function.

#### 3.4.2. Naive Bayes

Naive Bayes is the second learning model used in the proposed ensemble system. The Bayesian method is simply a method of classifying phenomena based on their likelihood of occurrence or non-occurrence. The probability of an event occurring in the future can be deduced from previous events of that event in the Naive bayes classification. Bayesian classification is used to solve problems in which each instance of *x* is chosen from a set of attribute values and the objective function *f* (*x*). By having the $<a_{1},a_{2},\ldots ,a_{n}>$ attribute values that describe the new sample, the Bayesian mechanism for classifying a new sample is to identify the most likely class or target value of *V_MAP_*. To classify the features in the proposed method, the simple Bayesian mechanism based on [29] is used.

#### 3.4.3. K-Nearest Neighbor

The K-nearest neighbor method is one of the simplest machine learning algorithms. This algorithm categorizes a sample based on a majority vote of its neighbors, and this sample is determined to be in the most general class among *k* close neighbors. Since it is effective, non-parametric, and simple to implement, the KNN method is applicable to a wide range of problems. As a result, the proposed method uses this classifier as one of the learning models in the ensemble system. In KNN, vectors in multidimensional feature space are used to represent training samples. The space is divided into sections with training samples. A point in space belongs to a class in which the majority of the training points within the *k* closest instances belong to that class [30]. The Euclidean distance criterion is used in the KNN model of the proposed method. In addition, the *k* parameter, or the number of nearest neighbors, is set to 3.

The voting technique is used as the final step in the proposed method for diagnosing a tumor. The voting technique’s goal is to improve the classification accuracy of the algorithms in comparison to the case, where each algorithm is used separately. Each classification algorithm may make mistakes when classifying some samples. The goal of voting-based techniques is to reduce resulting errors and increase accuracy. As a result, in the final step of the proposed method, the SVM, NB, and KNN classification models perform the classification operations of the test samples separately, and finally, the final output of the system is determined by voting on the results of all three models.

## 4. Results

The proposed model was implemented using MATLAB 2016a. A subset of the BRATS 2014 database [31] was used to evaluate the proposed method. There are 120 MRIs in this database, divided into two categories: normal (60 samples) and tumor (60 samples). MRIs of various slices of the brain are included in each sample in this database. An MRI slice is used to detect the presence of a tumor due to the large sample size. It is clear that this will not affect the generality of the proposed model’s application. The second database was Brain Tumor Detection 2020 (BTD20) [32]. This Database contains 3000 MRIs, out of which 1500 samples contain tumors, while the remaining 1500 samples are normal.

The images were prepared using ImageJ v1.49 software, which was used to extract and analyze MRIs from the database. This is an open-source application which converts raw data into image format. For this purpose, the information extracted from the database is stored in matrices with dimensions of 250×250.

The database samples were divided into two categories in the experiments: training samples and test samples. Due to small number of samples in the BRATS2014 database, the experiments were repeated 20 times in order to improve the validity of the test results. In total, 70% of the data was used to train the model in each iteration, with the remaining 30% used to test its performance. Training and test samples were chosen at random for each repetition. On the other hand, BTD20 includes a large number of samples. Thus, a 10-fold cross validation was performed to evaluate the performance of the proposed method in this database. In both cases, the number of image regions (number of thresholds *m*) was set to 5 in the SSO-based segmentation algorithm. These regions were used to separate each of the following regions on MRIs:Image background;The skull region;Regions of external tissue of the brain;Regions of internal tissue of the brain;The regions related to the tumor lesion.

Figure 3 shows an example of a segmented image created with the proposed SSO-based segmentation algorithm.

The SVD algorithm was used to extract the features of the segmented images. The appropriate number of features is equal to the number of features whose sum of squares of eigenvector variance is equal to 99 percent of the sum of squares of variance is the eigenvector of all features, as described in the previous section. Figure 4 shows the result of using the SVD algorithm for determining the number of extracted features. As shown in this diagram, the best number of features that can be extracted using the SVD algorithm is 43 (see Figure 4).

Figure 5 depicts the features extracted through the SVD algorithm from database samples as a colormap (Figure 5a) and a surface plot (Figure 5b).

In Figure 5a, features are displayed as columns of the feature matrix and samples are displayed as its rows. In this matrix, the lower half corresponds to features extracted from normal samples and the upper half corresponds to tumor samples. As it is clear in Figure 5, the first columns of the extracted feature matrix show more obvious differences between the two target classes, and this difference decreases as we progress through the extracted features. This attribute, shows that the feature extraction algorithm used in the proposed method can efficiently rank the features. Thus, the features that can show the difference between the target classes more clearly are prioritized. Figure 6 shows the distribution of samples belonging to each of the two target classes based on the first three features extracted by the SVD algorithm.

According to Figure 5 and Figure 6, and by examining the values of the extracted feature matrix for the two target classes, it can be seen that the samples of these two classes have recognizable differences, both in terms of the difference in the values of each feature and in terms of the probability of the presence of specific values for each feature (for example, the probability; the existence of a positive value for the first extracted feature is very low in normal samples and very high in tumor samples). Based on the first mentioned attribute (the difference between the corresponding features in each target class), distance-based algorithms (such as SVM and KNN) can distinguish the samples of these two categories with high accuracy; while the second mentioned attribute (probability of certain values for the corresponding features in each target class) is more suitable for use by probability-based classifiers (such as Naive Bayes). These findings show that by combining the learning models used in the proposed ensemble system, the extracted features can be classified with high accuracy.

The results of the proposed model’s correct diagnosis for 20 repetitions of experiments are shown in Figure 7 and Table 3. In these results, the proposed ensemble system’s accuracy is compared to each of the algorithms used in it.

Figure 7 compares the accuracy of the proposed ensemble system with the accuracy of the classifiers used for constructing it. As these results show, the proposed ensemble system can improve the diagnosis accuracy compared to the case that each constructing classifier is used separately. These results prove that the voting technique in ensemble systems is an efficient mechanism for improving the accuracy of weak classifiers.

Table 3, displays the average accuracy of correct diagnosis after 20 test iterations for BRATS2014 database. Compared to other cases, the proposed method can improve the accuracy of brain tumor diagnosis. The results of this experiment show that in addition to having a higher average accuracy, the proposed method has a smaller range of accuracy variations during different iterations. Figure 8, depicts the mean detection accuracy of the proposed ensemble system and the classification algorithms employed, as well as the accuracy change intervals. The upper and lower bounds of algorithmic accuracy changes at different iterations are represented by the thin lines of each box in this diagram.

As shown in Figure 8, the proposed method has the advantage of having higher and tighter bounds of accuracy change during different iterations. These findings show that in more than $\frac{3}{4}$ cases, the accuracy of the proposed method was greater than 97 percent, and in other cases, its accuracy was between 94 and 95 percent. In the proposed model, Figure 9 illustrates the confusion matrix resulting from brain tumor diagnosis in the BRATS 2014 database. The row/column index 1 in this matrix denotes the category of normal samples, while the row/column index 2 denotes the category of tumor samples. The proposed method correctly classified 97.6% of normal samples (365 samples out of 374 normal test samples), as shown in this matrix.

On the other hand, we were able to correctly identify up to 99.7 tumor samples (345 out of 346 tumor test samples) using the proposed ensemble system, with only one tumor sample misclassified. These findings show that the proposed method performs well in diagnosing healthy and tumor samples on average, correctly classifying 98.6% of the samples.

Figure 10 also shows the results of the confusion matrix for the other three compared algorithms.

Figure 11 compares the Receiver Operating Characteristic (ROC) curve of the proposed ensemble system with its constructing individual classifiers. The ROC curve demonstrates the achieved true positive rates versus false positive rates of classifiers in diagnosing brain tumors.

As shown in Figure 10, the ROC curve resulting from the proposed method results in higher TPR values and at the same time lower FPR values. These results show that by using the proposed method, fewer normal samples have been incorrectly diagnosed as tumor samples, and at the same time, more tumor samples have been correctly identified by the proposed method. Thus, the probability of correct diagnosis in the proposed method for the samples that are classified as cancerous is higher than the compared methods. The higher level of the ROC curve and more Area Under Curve (AUC) in Figure 10 confirms this point. Thus, the proposed ensemble system is effective in increasing the accuracy of classifiers in the diagnosis of tumors in brain MRIs.

The test results for the proposed algorithm for brain tumor diagnosis are shown in Table 3. The sensitivity and specificity criteria are compared in this table. The sensitivity criterion is calculated by dividing the total number of correctly identified tumor samples by the total number of tumor samples:(7)$$\mathrm{Sensitivity}=\frac{TP}{TP+FN}$$
where *TP* is the number of samples with brain tumors that have been correctly diagnosed, and *FN* is the number of samples with brain tumors that have been classified as normal. The specificity criterion is used to measure correctly classified normal samples. The following equation is used to calculate this criterion:(8)$$\mathrm{Specificity}=\frac{TN}{TN+FP}$$
where *TN* refers to the number of normal samples that have been correctly classified, and *FP* is the number of normal samples that have been incorrectly classified as tumor samples. The results presented in Table 4 are the average of 20 iterations. During these iterations, the same training and testing samples have been used for all algorithms. As the results of Table 4 show, the proposed method achieves a better performance than the compared algorithms, both in terms of correct detection percentage and in terms of sensitivity and specificity criteria. This higher efficiency can be attributed to the use of the ensemble technique in classifying the extracted features. By using multiple learning models, the accuracy and efficiency of learning algorithms can be increased, compared to the case where each one is used separately. On the other hand, using the combination of SSO algorithm (for segmentation of brain regions) and SVD algorithm (for extracting the features of segmented MRIs) has ensured that the proposed method is capable of describing the features of each MRI in a compact manner and showing the differences between the features of normal and tumor MRIs more clearly. The result of this is the better performance of the proposed method compared to previous methods such as [14,21,22,23]. These results confirm that the proposed method can be used as an effective tool for the automatic diagnosis of brain tumors in MRIs (see Table 4).

The proposed method was also evaluated by the BTD20 database. The results related to this experiment are presented in Table 5. These results, in addition to showing the overall superiority of the accuracy of the proposed method over the previous methods, confirm again that the accuracy of simple ML models in brain tumor diagnosis can be improved by using the ensemble learning strategy.

## 5. Discussion

The results of the experiments showed that using the proposed method, brain tumors can be detected in MRI images with an accuracy of about 98.61%, which is higher than the previous methods. This superior performance in the proposed method can be seen as the result of using two components: first, the use of SSO-based segmentation and SVD algorithm has made it possible to describe the characteristics of each MRI in a compact and efficient manner. Using the proposed segmentation method, the tumor region can be well separated from the others, and this is effective in reducing the complexity of the tumor diagnosis problem. On the other hand, as it was shown, by using the SVD algorithm, it is possible to extract the descriptive features of the image regions in such a way that in addition to increasing the processing speed (due to reducing the number of features), it is also effective in increasing the recognition accuracy (due to better representation of the difference between samples of two target classes). Second, using the ensemble technique in the classification phase of the proposed method has made it possible to take advantage of several classifiers at the same time. As the results presented in the previous section showed, the use of this strategy increases the detection accuracy of the proposed system compared to the case where each learning model is used separately (see Table 3, Table 4 and Table 5). However, achieving higher accuracy in the proposed method comes at the cost of increasing the time and processing resources required to train learning models. Although this increase in processing time is only evident in the training phase of the proposed model, this time difference can be minimized by using parallel processing techniques.

## 6. Conclusions

In this paper, data mining and machine learning techniques were used to propose a new method for diagnosing brain tumors through MRIs. The proposed method uses SSO to introduce a brain segmentation algorithm using the multilevel thresholding technique. This segmentation algorithm allows for a more precise identification of the tumor region in MRIs. The proposed method also uses SVD for feature extraction, which leads to a more compact and efficient mechanism for describing MRI features. Using SVD will increase the processing speed of features in the classification phase, in addition to removing redundant information which are not related to the existence of brain tumors in MRIs. In the proposed method, a combination of three classifiers as an ensemble model is used for diagnosis. The proposed combination is effective in covering the classification error of individual learners by the voting technique. The results indicate that the proposed method can diagnose brain tumors in the BRATS 2014 dataset with an average accuracy of 98.61%, sensitivity of 95.79% and specificity of 99.71%. On the other hand, the proposed method could diagnose brain tumors in the BTD20 database with an average accuracy of 99.13%, sensitivity of 99% and specificity of 99.26%. These results show a significant improvement compared to previous efforts. The findings confirm that using the image segmentation technique, as well as the ensemble learning, is effective in improving the efficiency of the proposed method.

One of the limitations of the proposed method is its longer training time, compared to the case when individual classifiers are used. Although, this time difference is almost unnoticeable in the test phase of the proposed method, it can be reduced using parallel processing techniques. Additionally, it is suggested to investigate the performance of other combinations of classifiers in the proposed ensemble system. Thus, ensemble systems based on other classifiers such as decision trees, artificial neural networks, random forests, and so on, can be investigated in future research. The proposed method may be effective in solving similar problems such as diagnosing Multiple Sclerosis (MS) disease. In this case, the problem of detecting brain tumors is translated to detecting MS plaques in brain MRIs. The authors believe more research in this area is crucial.

## Figures and Tables

**Figure 1 diagnostics-13-00561-f001:**
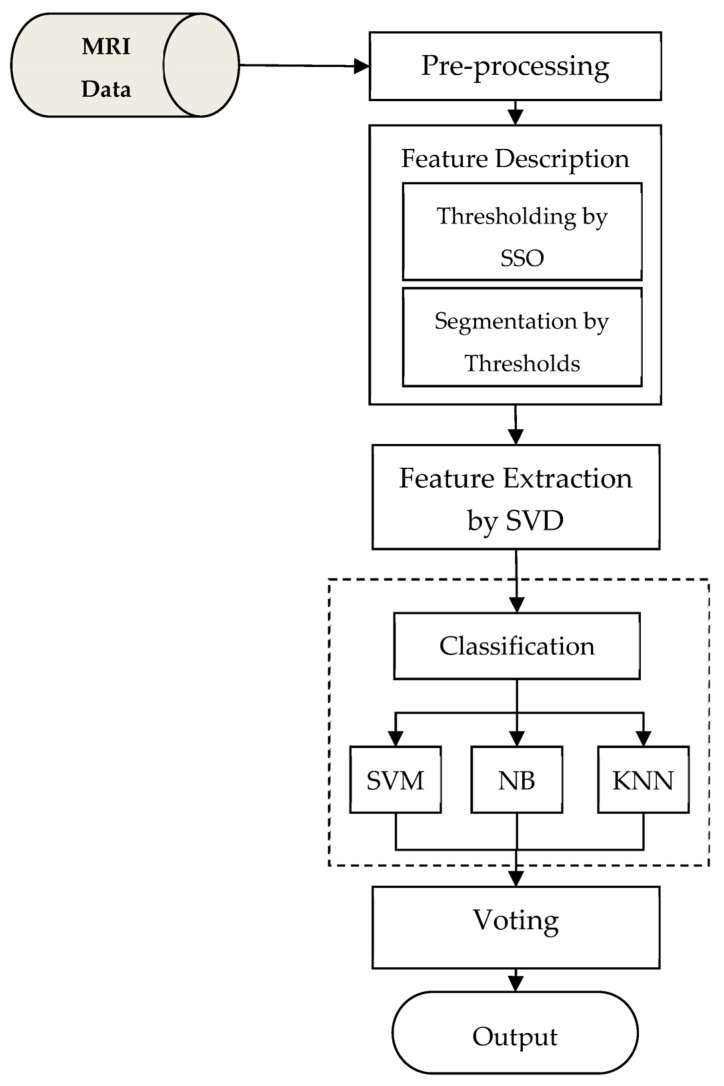
Block diagram of the proposed method.

**Figure 2 diagnostics-13-00561-f002:**
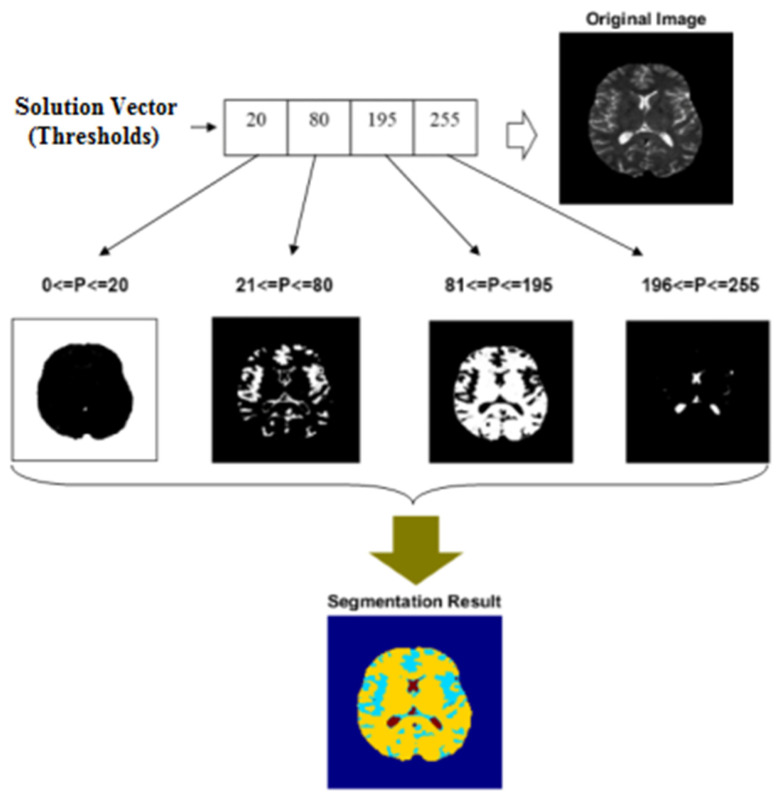
The structure of a solution vector in SSO algorithm for MRI segmentation.

**Figure 3 diagnostics-13-00561-f003:**
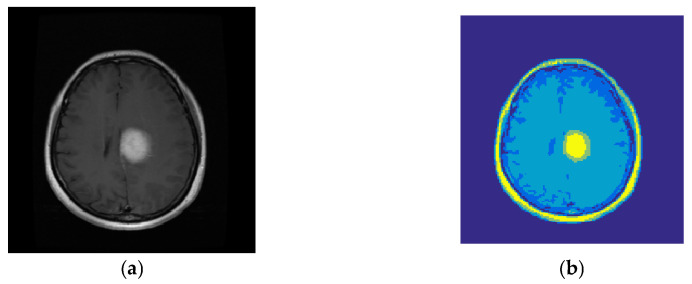
Result of image segmentation by the proposed method (**a**) A sample MRI from BRATS2014 (**b**) Result of segmentation.

**Figure 4 diagnostics-13-00561-f004:**
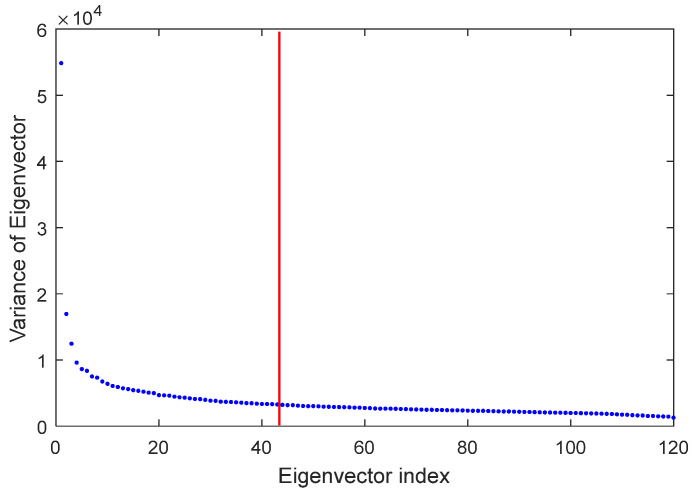
Determining the appropriate number of features in the SVD algorithm.

**Figure 5 diagnostics-13-00561-f005:**
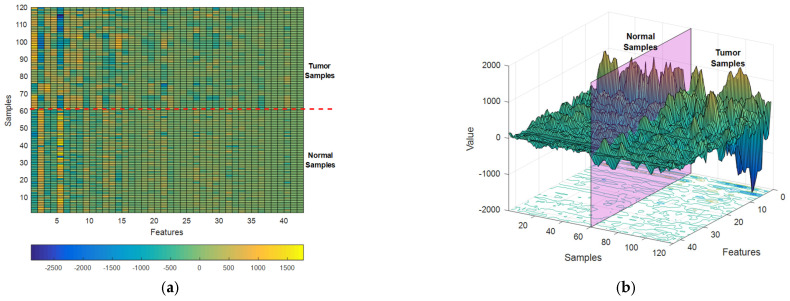
(**a**) The colormap, and (**b**) the surface plot of the features extracted through SVD algorithm.

**Figure 6 diagnostics-13-00561-f006:**
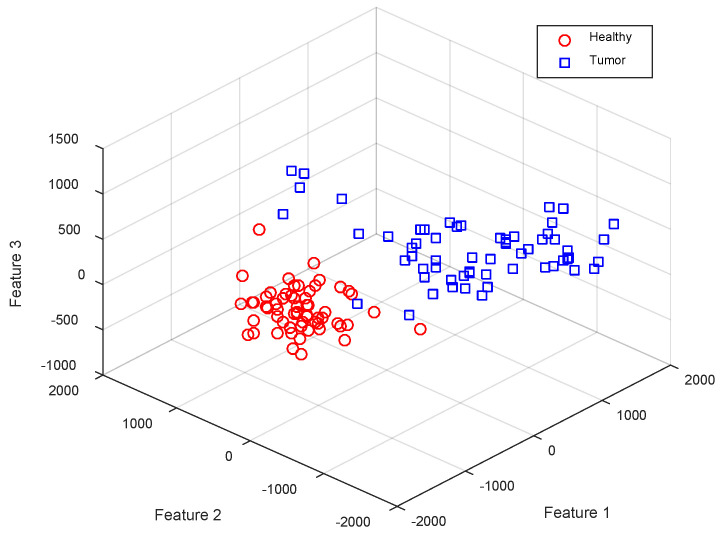
The distribution of samples belonging to each target class based on 3 features extracted by the SVD algorithm in BRATS2014.

**Figure 7 diagnostics-13-00561-f007:**
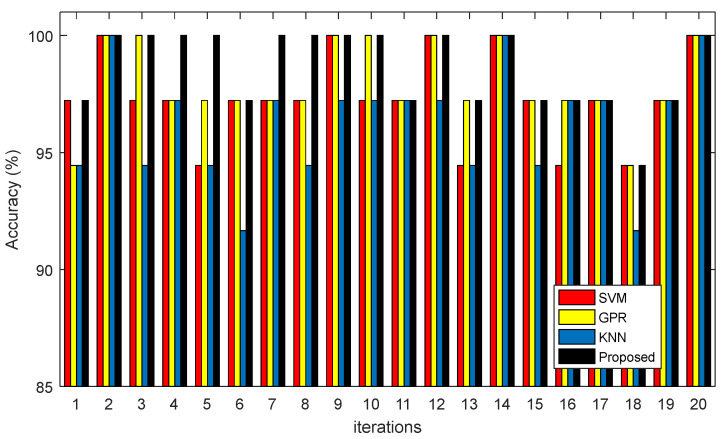
The accuracy of proposed method and other algorithms for 20 repetitions of experiments.

**Figure 8 diagnostics-13-00561-f008:**
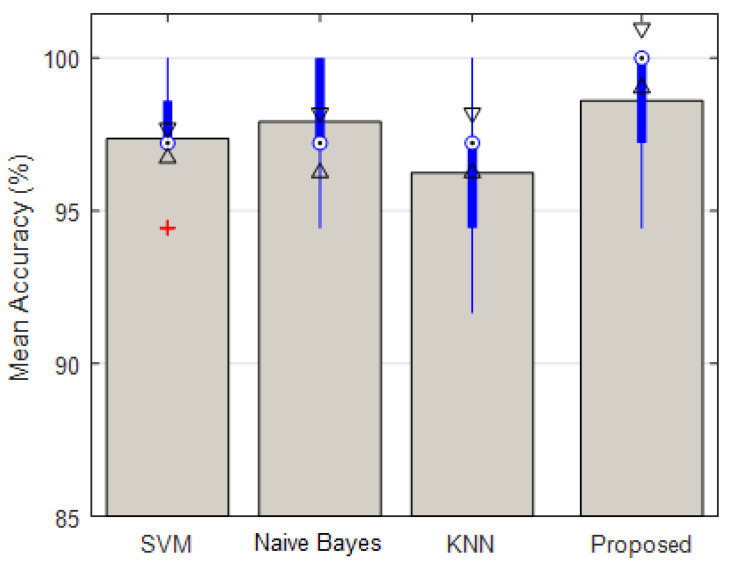
Average and boxplot of accuracy for different algorithms during 20 repetitions of experiments.

**Figure 9 diagnostics-13-00561-f009:**
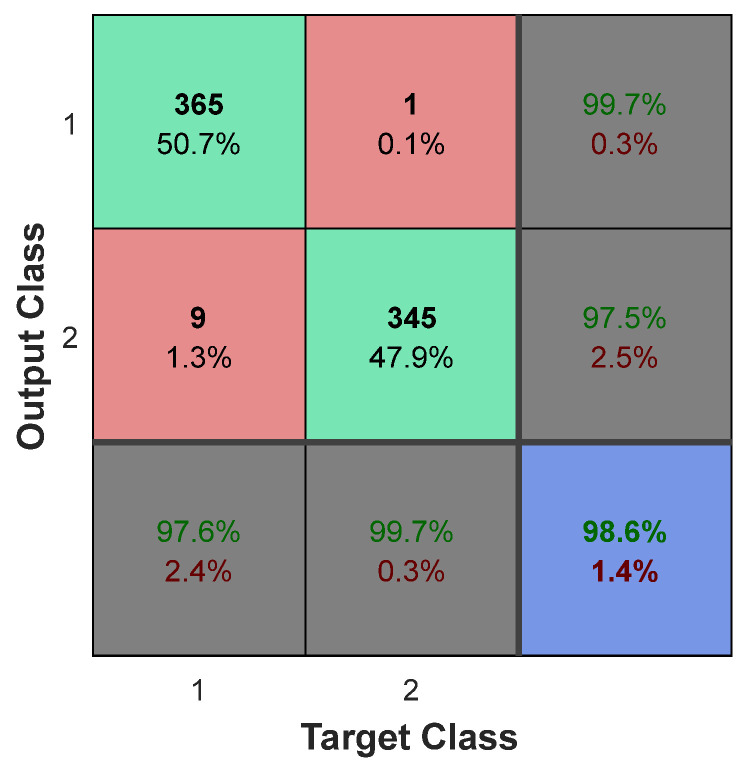
The confusion matrix of the proposed method as a result of brain tumor diagnosis in BRATS2014 database.

**Figure 10 diagnostics-13-00561-f010:**
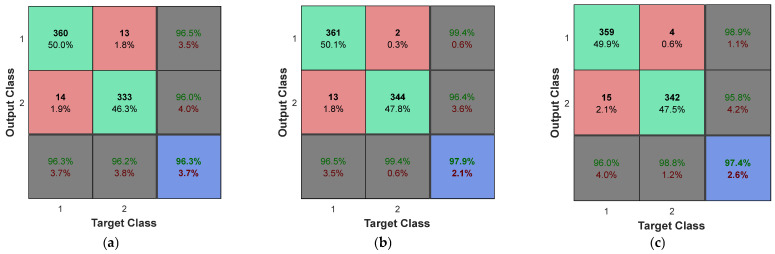
The confusion matrices of (**a**) KNN, (**b**) Naive Bayes, and (**c**) SVM as the results of brain tumor diagnosis in BRATS2014 database.

**Figure 11 diagnostics-13-00561-f011:**
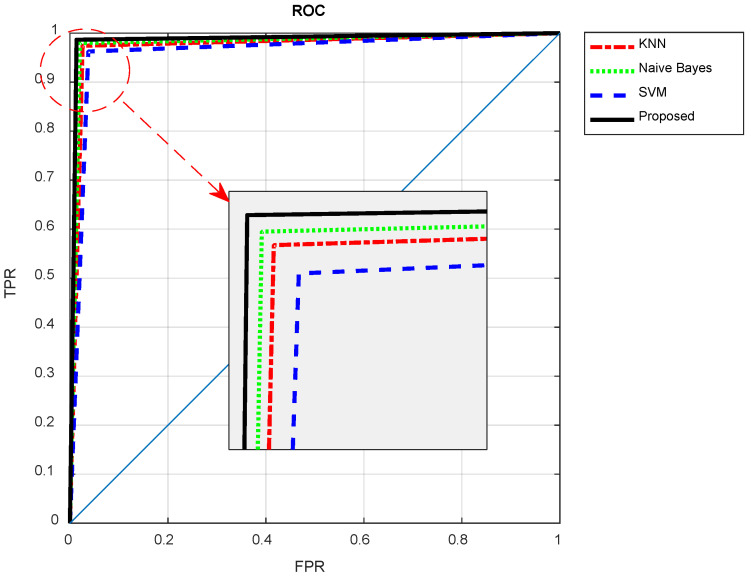
The ROC curve of the proposed method compared to other algorithms.

**Table 1 diagnostics-13-00561-t001:** Summary of the related works.

Author(s)	Year	Preprocessing/Feature Extraction Method	Classifier
Shobana & Balakrishnan [4]	2015	entropy and image energy	PNN
Shanmugapriya et al. [5]	2014	density and severity	SVM
Singh [6]	2016	Fuzzy segmentation	SVM
Nandpuru et al. [7]	2014	image normalization	SVM
Havaei et al. [12]	2017	deep features	DNN
Thaha et al. [13]	2019	image normalization	CNN + bat search
Rammurthy & Mahesh [14]	2020	image normalization	CNN + WHHO
Kang et al. [15]	2021	deep features of 3 CNNs	SVM
Ullah et al. [16]	2020	Median filter and HE	DNN
Ullah et al. [17]	2022	image normalization	MRA-UNet
Almalki et al. [18]	2022	deep features of CNN	SVM
Alsubai et al. [19]	2022	image normalization	CNN + LSTM
Alanazi et al. [20]	2022	binary classifiers of CNNs	22-layered CNN
Arif et al. [21]	2022	GLCM + GA	CNN
Arif et al. [22]	2022	GA + DWT + PSO	U-Net
Haq et al. [23]	2022	deep features of CNN	SVM

**Table 2 diagnostics-13-00561-t002:** Notations used in this paper.

Symbol	Description
$B_{i,j}$	The value in row *i* and column *j* of the black and white (binary) matrix *B*
$I_{i,j}$	Intensity value of the pixel in row *i* and column *j* of the input image *I*
$S_{q}$	Tsallis entropy criterion for evaluating the fitness of a solution in SSO algorithm
*C*	Number of target segments in SSO algorithm
*F*	The feature set, extracted by singular value decomposition
*FN*	The Number of false negative samples
*FP*	The Number of false positive samples
*m*	The number of segmentation thresholds in SSO algorithm
*p_i_*	The probability of value *i* in image
*q*	Quantization level of Tsallis entropy criterion
*T_i_*	The intensity threshold pairs [*t_i_*, *t_i+1_*) for *i*-th segmented region
*TN*	The Number of true negative samples
*TP*	The Number of true positive samples
*U*	The orthogonal matrix obtained by singular value decomposition
*V*	Transpose of the orthogonal matrix obtained by singular value decomposition
*Σ*	The diagonal matrix obtained by singular value decomposition
*FG*	The foreground area of the input image

The steps of the proposed method will be described in detail in the following sections.

**Table 3 diagnostics-13-00561-t003:** Summary of the detection results of the proposed method and each of the algorithms used in it.

Standard Deviation	Maximum Accuracy	Minimum Accuracy	Mean Accuracy	Title
**1.68**	**100**	**94.44**	**98.61**	Proposed method
1.77	**100**	**94.44**	97.92	Naive Bayes
2.01	**100**	**94.44**	97.36	SVM
2.43	**100**	91.67	96.25	KNN

**Table 4 diagnostics-13-00561-t004:** Comparison of the proposed method with previous works in diagnosis of brain tumors in BRATS2014 database.

Algorithm	Specificity (%)	Sensitivity (%)	Accuracy (%)
proposed method	**99.7110**	97.5936	**98.6111**
Naïve Bayes	99.4220	96.5241	97.9167
SVM	98.8439	95.9893	97.3611
KNN	96.2428	96.2567	96.2500
Rammurthy et al. [14]	97.4010	78.0000	91.1600
Arif et al. [21]	97.0000	**98.6000**	98.5000
Arif et al. [22]	98.0000	98.0000	97.0000
Haq et al. [23]	98.1256	97.3105	98.3000

**Table 5 diagnostics-13-00561-t005:** Comparison of the proposed method with previous works in diagnosis of brain tumors in BTD20 database.

Algorithm	Specificity (%)	Sensitivity (%)	Accuracy (%)
proposed method	**99.2667**	**99.0000**	**99.1333**
Naive Bayes	**99.2667**	98.6667	98.9667
SVM	98.2000	98.6000	98.9000
KNN	98.9333	98.4000	98.6667
Kang et al. [15]	98.4100	98.9300	98.6700
Ullah et al. [16]	95.6500	96.0000	95.7700
Alsubai et al. [19]	98.8000	98.9000	99.1000

## Data Availability

Not applicable.

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
