# Peer review of "An Ensemble Model for the Diagnosis of Brain Tumors through MRIs"

_diagnostics, 2023, doi:10.3390/diagnostics13030561_

Round 1
Reviewer 1 Report
This paper presents an ensemble model for brain tumors diagnosis using MRIs. The experimental study is interesting information in this paper. However, the main weakness of the paper lies in its lack of originality and novelty. The following suggestions may be considered to enhance the quality and clarity of the manuscript.
1) The motivation is not clear. Why did this work? Is any problem does it address that the previous method cannot?
2) This area is rapidly evolving, and new papers have been published. Therefore, some state-of-the-art papers should be taken into account are:
https://www.mdpi.com/1424-8220/21/6/2222
https://www.sciencedirect.com/science/article/abs/pii/S0306987720308689
https://www.sciencedirect.com/science/article/pii/S0020025522007332
3) In the related works, existing studies can also be summarized in a tabular form to improve readability
4) In experiments, the experimental analysis seems insufficient, which can not verify the motivation and contributions of this work.
5) Frankly, using machine learning approaches for brain tumor classification and segmentation is not quite new, e.g., multiple other approaches have been used for comparison by the authors in the case study part. Then, what is revolutionarily new about this study that uses an ensemble model for the diagnosis of brain tumors?
6) How to deal with overfitting in your model?
7) The best results should be bolded in the table
8) There are many typos and errors
9) Besides language also needs improvement
Author Response
Reviewer #1
Open Review
( ) I would not like to sign my review report
(x) I would like to sign my review report
English language and style
( ) English very difficult to understand/incomprehensible
(x) Extensive editing of English language and style required
( ) Moderate English changes required
( ) English language and style are fine/minor spell check required
( ) I don't feel qualified to judge about the English language and style
|
Yes |
Can be improved |
Must be improved |
Not applicable |
|
|
Does the introduction provide sufficient background and include all relevant references? |
( ) |
(x) |
( ) |
( ) |
|
Are all the cited references relevant to the research? |
( ) |
(x) |
( ) |
( ) |
|
Is the research design appropriate? |
( ) |
(x) |
( ) |
( ) |
|
Are the methods adequately described? |
( ) |
( ) |
(x) |
( ) |
|
Are the results clearly presented? |
( ) |
( ) |
(x) |
( ) |
|
Are the conclusions supported by the results? |
( ) |
(x) |
( ) |
( ) |
Comments and Suggestions for Authors
This paper presents an ensemble model for brain tumors diagnosis using MRIs. The experimental study is interesting information in this paper. However, the main weakness of the paper lies in its lack of originality and novelty. The following suggestions may be considered to enhance the quality and clarity of the manuscript.
1) The motivation is not clear. Why did this work? Is any problem does it address that the previous method cannot?
The authors tried to clarify the motivation of the work. The content related to this case were highlighted in Introduction section.
2) This area is rapidly evolving, and new papers have been published. Therefore, some state-of-the-art papers should be taken into account are:
The authors appreciate reviewer for the suggested works. These papers, in addition to several recent works were reviewed in section 2. The content related to this case were highlighted in section 2.
3) In the related works, existing studies can also be summarized in a tabular form to improve readability
The studied works were summarized in Table 1. The content related to this case were highlighted in section 2.
4) In experiments, the experimental analysis seems insufficient, which can not verify the motivation and contributions of this work.
The authors, tried to enrich the analysis and results section by performing experiments on a new and larger database. The results related to this case were highlighted in section 4.
5) Frankly, using machine learning approaches for brain tumor classification and segmentation is not quite new, e.g., multiple other approaches have been used for comparison by the authors in the case study part. Then, what is revolutionarily new about this study that uses an ensemble model for the diagnosis of brain tumors?
The authors attempted to clarify the novelty of the work in introduction section. The content related to this case were highlighted in Introduction section.
6) How to deal with overfitting in your model?
This case was described and highlighted in section 3 (lines 217 to 222).
7) The best results should be bolded in the table
The best results were bolded in all tables.
8) There are many typos and errors
The manuscript was double-checked for typos and errors.
9) Besides language also needs improvement
The manuscript was double-checked for grammar mistakes and errors were corrected.

Reviewer 2 Report
In this paper, the authors propose an ensemble of different classifiers for enhancing the classification accuracy of the brain tumor classification. The paper needs thorough revision to clearly explain the findings and improvements made to the literature. The following are the comments.
1) The section Related Works does not present the importance of deep learning in brain tumor classification. There are several related works using deep learning. However, this section lacks in describing the importance of deep learning. Some of the papers are listed below; however, there are several other recent papers as well. The authors should strengthen their literature review by including the importance of deep learning in the related works.
https://doi.org/10.3390/diagnostics12081793
https://doi.org/10.3389/fncom.2022.1005617
https://doi.org/10.3390/s22010372
2) Figure 1: Text in some blocks is not readable.
3) Table (1) should be replaced with Table 1 in the text. same for other tables as well. Similarly, for figures
4) How voting criteria is implemented. There is no information about the voting criteria or discussion of its results in the manuscript.
5) In the confusion matrix of Figure 8, there are 366 samples for tumors and 354 for normal, whereas the dataset contains 120 MRI images. Also, in Figures 8 and 9, the total sample size is not the same, why? More explanation is needed to further elaborate on the generation of confusion matrices. I am not convinced by these results.
6) Further detail is also required for the training and testing procedures. Is any cross-validation used?
Author Response
Reviewer #2
Open Review
(x) I would not like to sign my review report
( ) I would like to sign my review report
English language and style
( ) English very difficult to understand/incomprehensible
( ) Extensive editing of English language and style required
( ) Moderate English changes required
(x) English language and style are fine/minor spell check required
( ) I don't feel qualified to judge about the English language and style
|
Yes |
Can be improved |
Must be improved |
Not applicable |
|
|
Does the introduction provide sufficient background and include all relevant references? |
( ) |
(x) |
( ) |
( ) |
|
Are all the cited references relevant to the research? |
( ) |
( ) |
(x) |
( ) |
|
Is the research design appropriate? |
( ) |
( ) |
(x) |
( ) |
|
Are the methods adequately described? |
( ) |
( ) |
(x) |
( ) |
|
Are the results clearly presented? |
( ) |
( ) |
(x) |
( ) |
|
Are the conclusions supported by the results? |
( ) |
( ) |
(x) |
( ) |
Comments and Suggestions for Authors
In this paper, the authors propose an ensemble of different classifiers for enhancing the classification accuracy of the brain tumor classification. The paper needs thorough revision to clearly explain the findings and improvements made to the literature. The following are the comments.
1) The section Related Works does not present the importance of deep learning in brain tumor classification. There are several related works using deep learning. However, this section lacks in describing the importance of deep learning. Some of the papers are listed below; however, there are several other recent papers as well. The authors should strengthen their literature review by including the importance of deep learning in the related works.
The authors appreciate reviewer for the suggested works. These papers, in addition to several recent works were reviewed in section 2. The content related to this case were highlighted in section 2.
2) Figure 1: Text in some blocks is not readable.
Shapes and font size in Figure 1 were corrected to make the contents readable.
3) Table (1) should be replaced with Table 1 in the text. same for other tables as well. Similarly, for figures
All Table and Figure numbers were corrected, according to the format.
4) How voting criteria is implemented. There is no information about the voting criteria or discussion of its results in the manuscript.
The voting mechanism was described in final paragraph of section 3 (Lines 399 to 406). Also, the effect of voting on accuracy of the proposed method was discussed in section 4. To do this, the authors compared the performance of voting mechanism with the case that each classifier diagnoses brain tumor individually.
5) In the confusion matrix of Figure 8, there are 366 samples for tumors and 354 for normal, whereas the dataset contains 120 MRI images. Also, in Figures 8 and 9, the total sample size is not the same, why? More explanation is needed to further elaborate on the generation of confusion matrices. I am not convinced by these results.
In experiments, related to BRATS2014 database, we performed training-testing operation 20 times. During each iteration, 30% of samples were randomly selected for testing (36 samples). Thus, after 20 iteration, the number of test samples was 720 which is equal with the sum of confusion matrices in figures 8 and 9. In these matrices, there are 374 test samples which belong to normal class (sum of first column) and 346 samples belonging to tumor class (sum of second column). In these matrices, columns represent the target classes, while rows represent the outputs.
6) Further detail is also required for the training and testing procedures. Is any cross-validation used?
The training and testing procedures were explained in the third paragraph of section 4 (lines 420 to 427).

Reviewer 3 Report
The submitted manuscript proposed an ensemble model for diagnosis of brain tumors in MRI images based on the social spider optimization (SSO) algorithm and the SVD technique. By combining these techniques, the authors tried to provide a segmentation algorithm that aims to allow for a more precise identification of the tumor region and to speed up the classification.
However, the authors have not successfully addressed the following issues:
1. The model is tested for BRATS 2014 dataset only. It is hard to know if the proposed method will work for general MRI images.
2. In fact, the manuscript does not explain what types of brain MRI images (contrast, resolutions, etc.) the model is targeted for. As brain MR images have different characteristics depending on what parameters (and MRI scanners) they use, it is hard to know if the proposed algorithm is actually applicable for real MRI images.
3. The methods that the proposed method is compared with are not up to date. Recently, many analytic and deep-learning based segmentation and classification methods are in use. To validate the superiority of the proposed method, the propoed method should be compared with more recent algorithms.
In general, the algorithm and the manuscript should be largely improved. For example, some of the thresholding values are heuristically defined without explaining the reasoning behind. Only a few images for segmentation and classification results are provided, which makes it difficult to assess the quality of the proposed method.
Author Response
Reviewer #3
Open Review
(x) I would not like to sign my review report
( ) I would like to sign my review report
English language and style
( ) English very difficult to understand/incomprehensible
(x) Extensive editing of English language and style required
( ) Moderate English changes required
( ) English language and style are fine/minor spell check required
( ) I don't feel qualified to judge about the English language and style
|
Yes |
Can be improved |
Must be improved |
Not applicable |
|
|
Does the introduction provide sufficient background and include all relevant references? |
( ) |
( ) |
(x) |
( ) |
|
Are all the cited references relevant to the research? |
( ) |
( ) |
(x) |
( ) |
|
Is the research design appropriate? |
( ) |
( ) |
(x) |
( ) |
|
Are the methods adequately described? |
( ) |
( ) |
(x) |
( ) |
|
Are the results clearly presented? |
( ) |
( ) |
(x) |
( ) |
|
Are the conclusions supported by the results? |
( ) |
( ) |
(x) |
( ) |
Comments and Suggestions for Authors
The submitted manuscript proposed an ensemble model for diagnosis of brain tumors in MRI images based on the social spider optimization (SSO) algorithm and the SVD technique. By combining these techniques, the authors tried to provide a segmentation algorithm that aims to allow for a more precise identification of the tumor region and to speed up the classification.
However, the authors have not successfully addressed the following issues:
- The model is tested for BRATS 2014 dataset only. It is hard to know if the proposed method will work for general MRI images.
The authors, tried to enrich the analysis and results section by performing experiments on a new and larger database. The results related to this case were highlighted in section 4.
- In fact, the manuscript does not explain what types of brain MRI images (contrast, resolutions, etc.) the model is targeted for. As brain MR images have different characteristics depending on what parameters (and MRI scanners) they use, it is hard to know if the proposed algorithm is actually applicable for real MRI images.
The proposed model targets the contrast/intensity attributes of MRIs. All processing steps of the proposed method work on contrast and intensity values of the images.
- The methods that the proposed method is compared with are not up to date. Recently, many analytic and deep-learning based segmentation and classification methods are in use. To validate the superiority of the proposed method, the propoed method should be compared with more recent algorithms.
The results of diagnosis by proposed method were compared with recent works and results presented in Tables 4 and 5.
- In general, the algorithm and the manuscript should be largely improved. For example, some of the thresholding values are heuristically defined without explaining the reasoning behind. Only a few images for segmentation and classification results are provided, which makes it difficult to assess the quality of the proposed method.
The authors thank the reviewer for these useful comments. We tried to improve the manuscript by clarifying the mentioned problems. For example, the authors stated the reasoning behind choosing the thresholds of 0.05 and 99% for preprocessing and feature extraction steps, respectively. Also, new figures were added to the results section in order to improve the quality of the work. These content were highlighted in the manuscript.

Round 2
Reviewer 2 Report
The authors have addressed my points raised in the initial version of the manuscript.
Author Response
The authors have addressed my points raised in the initial version of the manuscript?
The authors attempted to resolve the reviewer’s concerns in the revised version of the manuscript. These cases include the following:
- Related works: the suggested papers were reviewed in related works section. Also, several recent works were added to this section.
- Figure 1: the blocks of this figure were edited, in order to make the contents readable.
- Table and Figure numbers: the format of referring to Tables and Figures in the text were corrected as the reviewer ordered.
- The voting criteria: as stated in the final paragraph of section 3, the proposed ensemble system uses majority voting mechanism for diagnosis. In this mechanism, each test sample belongs to the class which at least two classifiers have voted for that class (produced the output label of that class).
- The numbers in the confusion matrix: The confusion matrix in figure 8, is the classification results of proposed method for BRATS2014 database after 20 iterations. In each iteration of this experiment, 70% of database samples were randomly selected for training, while the remaining 30% of samples (36 samples) were used for testing. This means that after 20 iterations, 720 test samples were classified. Since the selection of samples were done randomly, thus it is obvious that the number of samples belonging to each class are not same as the original database. The authors tried to explain this case more clearly in previous response letter.
- Training and testing procedures: The training and testing procedures were explained in the third paragraph of section 4 (lines 420 to 427). As stated there, in BRATS2014 database, the experiments were repeated 20 times in order to improve the validity of the test results. 70% of the samples were chosen randomly to train the model in each iteration, and the remaining 30% samples were used to test its performance. On the other hand, for BTD20 database, a 10-fold cross validation experiment was performed. This means 10 iterations of training-testing operations. Thus the database was partitioned into 10 parts (without common instances). In each iteration a new part was used for testing, while remaining 9 parts were used for training ML models.

Reviewer 3 Report
The manuscript is generally improved but my concerns are not clearly resolved in the revised form, especially regarding the specific target of the proposed work. i.e., for what specific target images the proposed method is useful. In general, the novelty and originality are weak.
Author Response
The manuscript is generally improved but my concerns are not clearly resolved in the revised form, especially regarding the specific target of the proposed work. i.e., for what specific target images the proposed method is useful. In general, the novelty and originality are weak.
The authors are doubtful about clear understanding of the issue raised by the reviewer about targeted MRI types. In this research, T1-weighted brain MRIs were mainly used for diagnosis. In this type of MRIs, the signal of the fatty tissue is enhanced (shown as white) and the signal of the water is suppressed (shown as gray). The proposed method uses contrast and intensity features of these images for diagnosis (as stated in segmentation step, which is done by SSO and multi-level thresholding of intensity values) and it is not dependent on the resolution of the images.
Although the proposed method does not use completely novel techniques for diagnosis, but it has used the existing techniques in a new manner, which have not been addressed in previous researches. There are many researches that have followed this strategy; For example: references [21], [22], [23] in the revised version of the manuscript. The proposed method uses SSO for initial MRI segmentation and uses SVD for extracting features from segmented images. To the best of our knowledge, this combination has not been used in brain tumor diagnosis applications. On the other hand, the combination of ML models in proposed ensemble system is unique, in this research topic. Considering the obtained results, these techniques are effective in improving the diagnosis performance which may be interesting for research community.
